# CERTIFIED DEFENSE AGAINST COMPLEX ADVERSARIAL ATTACKS WITH DYNAMIC SMOOTHING

## ABSTRACT

Randomized smoothing has emerged as a certified defence mechanism with probabilistic guarantees that works at scale. However, current randomized smoothing methods offer theoretical guarantees that are limited by their reliance on specific noise distributions, and they struggle to handle complex adversarial attacks. In this paper, we propose a novel certification method based on randomized smoothing designed to handle complex adversarial attacks, including combinations of multiple attack types. We call this method Dynamic Smoothing (DSMOOTH). Our key idea is to incorporate more general distributions for smoothing then isotopic Gaussian noise, for which probabilistic guarantees can be derived in terms of the Mahalanobis distance. These general distributions make the smoothed classifier more robust against a wide range of threats, including localized adversarial attacks and multi-attacks. We validate the performance of our method experimentally on challenging threat models using CIFAR-10 and IMAGENET, and demonstrate its superiority over state-of-the-art defenses in terms of certified accuracy. Our results show that the proposed method significantly improves the robustness of machine learning models against complex attacks, advancing their suitability for use in safety-critical applications. Code: [removed for review]

## 1 INTRODUCTION

Machine Learning has seen considerable progress in recent years, especially with deep neural networks (DNNs). However, these networks are vulnerable to adversarial examples (Szegedy et al., 2013; Goodfellow et al., 2014; Zhao et al., 2023), posing a challenge for their use in safety-critical areas (Kurakin et al., 2016; Shayegani et al., 2023). Adversarial attacks, such as DeepFool (Moosavi-Dezfooli et al., 2016), AutoAttack (Croce & Hein, 2020), patch-based attacks (Brown et al., 2017b), and attacks on LLMs (Zou et al., 2023) continue to evolve, outpacing existing defenses and creating a persistent struggle between attackers and defenders (Carlini & Wagner, 2017a; Madry et al., 2017a). Current defenses, e.g., denoising generative models (Gu & Rigazio, 2014; Ho et al., 2020), adversarial training (Miller et al., 2020; Kireev et al., 2022), and defensive distillation (Papernot et al., 2016a; Wang et al., 2021), have not fully succeeded in preventing stronger attacks. Hence, the problem of building trustworthy ML systems suitable for critical applications remains an open question.

Certified robustness has emerged as an alternative approach, with randomized smoothing (Lecuyer et al., 2019; Li et al., 2019; Cohen et al., 2019b; Anderson & Sojoudi, 2022; Scholten et al., 2023; Anani et al., 2024) being a notable method. This technique, which provides *probabilistic* guarantees, involves creating a smoothed classifier by applying Gaussian noise to the base classifier. This method was shown by Lecuyer et al. (2019) and Li et al. (2019) to provide consistent classification within a certified radius under $\ell_2$ norm considerations, although the guarantees were initially loose. Cohen et al. (2019b) were the first to offer tight robustness guarantees for this method against $\ell_2$ norm-constrained adversarial attacks, sparking further studies in this area.

Randomized smoothing has become a widely recognized method for certified robustness, though it has limitations. Cohen et al. (2019b) identified the need for further exploration of $\ell_p$ norms beyond $\ell_2$. Recent works have been addressing robustness guarantees for randomized smoothing against various types of adversaries, including $\ell_1$-bounded attacks (Teng et al., 2020), $\ell_0$-bounded attacks (Levine & Feizi, 2020c; Lee et al., 2019), and Wasserstein attacks (Levine & Feizi, 2020a). However, defending against complex, high-dimensional adversarial attacks remains an open challenge.

**Our Contribution.**

- We provide a certification method based on randomized smoothing, which we refer to as **D**ynamic **SMOOTH**ing (DSMOOTH, Sec. 4.1). DSMOOTH uses more complex smoothing distributions than traditional randomized smoothing, making the smoothing process more adaptable to localized and non-uniform adversarial attacks then previous methods. Our method is also a suitable certified defense method against attacks based on multiple norms, such as multi-attacks.

- We derive probabilistic guarantees based on the Mahalanobis distance (Sec. 4.2). Our analysis, which is non-trivial, provides a framework to derive guarantees using push-forward measures (Thm. 4.4), which can be of independent interest. Furthermore, we derive probabilistic guarantees using the $\ell_2$ norm, recovering known guarantees for isotopic Gaussian noise (Cor. 4.7).

- We provide extensive experiments on CIFAR-10 and IMAGENET, considering a multi-attack that combines the Square Attack algorithm (Andriushchenko et al., 2020) and FGSM (Goodfellow et al., 2015). We show that DSMOOTH achieves good certified accuracy, significantly outperforming baselines (Sec. 5).

## 2 RELATED WORK

Since there is a large amount of scientific articles on this topic, we only discuss the contributions relevant for this work. The interested reader can refer to, e.g., Kumari et al. (2023); Kwiatkowska & Zhang (2023), for a more complete overview. Defenses against adversarial examples fall into empirical and certified categories. Empirical defenses, e.g., adversarial training (Madry et al., 2017a;b; Jin et al., 2023), aim to enhance robustness but lack guarantees of being unbreakable, as many have been compromised by stronger attacks, e.g., (Carlini & Wagner, 2017b; Athalye et al., 2018; Tramèr et al., 2020). Certified defenses and verification methods ensure consistent classifier output within a small neighborhood of $x$, using exact methods, e.g., (Huang et al., 2017; Katz et al., 2017; Ehlers, 2017; Mao et al., 2023; 2024), or conservative methods, e.g., (Wong & Kolter, 2018; Raghunathan et al., 2018; Dvijotham et al., 2018). Randomized smoothing has emerged as a probabilistic certified defense mechanism that works at scale.

Although the literature on randomized smoothing largely focuses on simple threat models, such as imperceptible adversarial perturbations of the input images (Szegedy et al., 2014; Goodfellow et al., 2015; Papernot et al., 2016b; Carlini & Wagner, 2017c), more complex threat models have been considered. Patch attacks, which place imperceptible modifications on images, can cause misclassifications and compromise system security. Levine & Feizi (2020b) address this with (De-) Randomized Smoothing for certifiable defense, leveraging the constraints of patch attacks over general sparse attacks. Zhang et al. (2023) introduce DRSM (De-randomized smoothed MalConv), adapting de-randomized smoothing for malware detection through executables (Raff et al., 2018). Recently, randomized smoothing has been used against image transformations (Fischer et al., 2020). Randomized smoothing has also been extended to discrete data (Bojchevski et al., 2020).

## 3 FRAMEWORK

### 3.1 PROBLEM DESCRIPTION

We are given a pre-trained classifier $f$. We do not make any specific assumption on the inner workings of $f$. For instance, $f$ can be a large convolutional neural network, e.g., ResNet (He et al., 2016), MobileNet (Howard et al., 2017), or any other model suitable for perception tasks in autonomous vehicles. We consider a threat model for the classifier $f$. This threat model generates adversarial images $\hat{x}$ by adding perturbations $\hat{\delta}$ to input images $x$, with the goal of fooling the classifier at inference time. In this work, we consider general *white-box* adversarial attacks, i.e., attacks in which the attacker may have full access to and knowledge of the target model's architecture, parameters, and training data. Formally, we consider the following class of adversarial attacks:

**Definition 3.1.** *Consider a classifier $f$ with a loss function $\mathcal{L}$. For an input $x$ with label $y$, a white-box attack for $f$ generates an adversarial example $\hat{x} = x + \hat{\delta}$, such that*

$$\hat{\delta} = \arg\max_{\delta \in \mathcal{C}(\delta)} \mathcal{L}(f(x + \delta), y). \tag{1}$$

*Here, $\mathcal{L}$ is a loss function and $\mathcal{C}(\delta)$ is a perturbation set, which is the collection of all possible perturbations $\delta$ that can be applied to an input $x$. A* white-box multi-attack *is an adversarial strategy that combines multiple white-box attacks as in equation 1 to generate a single adversarial example.*

Adversarial attacks as in equation 1 encompass a wide variety of attacks, such as spatial perturbations (Engstrom et al., 2019), Wasserstein- bounded perturbations (Hu et al., 2020; Wong et al., 2019), perturbations of the image colors (Laidlaw & Feizi, 2019) or perceptual adversarial attacks (Laidlaw et al., 2021; Wong & Kolter, 2021). Adversarial attacks on traffic sign detection by (Li et al., 2021) and physical adversarial attacks (Brown et al., 2017a; Woitschek & Schneider, 2023) are also attacks as in Def. 3.1.

Multi-attacks as in Def. 3.1 combine any of these methods, to exploit a broader range of model vulnerabilities. Multi-attacks optimize perturbations under different norms, leading to more complex, non-uniform perturbations. An example of a multi-attack, which is used for experimental comparison in this work, is a combination of the Square Attack algorithm (Andriushchenko et al., 2020) with FGSM (Goodfellow et al., 2015). This attack, which we denote as SQUARE + FGSM, first applies a Square Attack to an input image, and then it applies a FGSM attack to the resulting sample.

**The research question.** We study the problem of providing a certified defense mechanism against adversarial attacks as in Def. 3.1. This defense mechanism ought to be suitable to handle highly-dimensional input, such as images in datasets for vision-based perception systems of robots and autonomous driving systems.

## 3.2 RANDOMIZED SMOOTHING

Randomized smoothing is a technique for improving the robustness of models against adversarial attacks (Lecuyer et al., 2019; Li et al., 2019; Cohen et al., 2019b). The main principle of randomized smoothing is to transform a deterministic classifier into a probabilistic one by averaging its predictions over many noisy versions of the input. This process effectively "smooths out" the decision boundary of the classifier, making it less sensitive to input perturbations. Specifically, given a classifier $f$, randomized smoothing is a method for constructing a new classifier $g$ as

$$g(x) \coloneqq \arg\max_{y} \mathbb{P}\left(f(x + \varepsilon) = y\right) \quad \text{with} \quad \varepsilon \sim \mathbb{P}\left(\varepsilon\right).$$

Here, $\mathbb{P}\left(\varepsilon\right)$ is the *smoothing distribution* and it determines how noise is added to the input $x$. Typically, the smoothing distribution is a Gaussian distribution of the form $\mathbb{P}\left(\varepsilon\right) = \mathcal{N}(0, \sigma^2 I)$, with $I$ the identity matrix and $\sigma$ a user-defined scalar, although other distributions have been considered (see, e.g., (Teng et al., 2020; Levine & Feizi, 2020c; Lee et al., 2019)). Randomized smoothing provides probabilistic robustness guarantees in terms of the *certified radius*. This radius specifies a region around an input $x$ within which the smoothed classifier's prediction is guaranteed to be robust, with a certain probability. The region specified by the certified radius is called a *safety region*. The choice of the smoothing distribution significantly affects the robustness guarantees provided by randomized smoothing. The guarantees obtained with standard Gaussian smoothing distributions, as above, specify a safety region $\mathcal{S}$ using $\ell_p$ norms, e.g., $\mathcal{S} \coloneqq \{\hat{x} \colon \|\hat{x} - x\|_p \leq R\}$ for some radius $R$. These types of guarantees are suitable to certify robustness against imperceptible adversarial perturbations on the input image, such as those generated by L-BFGS (Szegedy et al., 2014), FGS (Goodfellow et al., 2015), DeepFool (Moosavi-Dezfooli et al., 2016), JSMA (Papernot et al., 2016b), or CW (Carlini & Wagner, 2017c). However, due to their reliance on global noise perturbations, guarantees based on isotopic Gaussian smoothing may be unsuitable for complex attacks that use structured and localized adversarial perturbations.

## 4 METHODOLOGY

### 4.1 OVERVIEW

We extend the randomized smoothing framework by Cohen et al. (2019a) to more complex smoothing distributions. In contrast to prior work, our framework uses *anisotopic* Gaussian noise as a smoothing distribution, i.e., a Gaussian distribution in which the variances along different dimensions of the space are not equal, which allows to handle both sparse and localized adversarial perturbations. Importantly, in Sec. 4.2 we derive probabilistic guarantees for this method that generalize previous known guarantees (Cohen et al., 2019a).

To define our smoothing framework, consider general adversarial examples of the form $\hat{x} = x + \hat{\delta}$ constructed with a general white-box (multi)attack as in Def. 3.1. We can view $\hat{\delta}$ as a random variable, where the randomness is given by the choice of the corresponding natural example $x$. We define the covariance matrix $\Sigma$ such that its entries are

$$[\Sigma]_{i,j} \coloneqq \text{Cov}[\hat{\delta}_i, \hat{\delta}_j] = \mathbb{E}[(\hat{\delta}_i - \mathbb{E}[(\hat{\delta}_i)])(\hat{\delta}_j - \mathbb{E}[(\hat{\delta}_j)])], \qquad (2)$$

with $\hat{\delta}_i$ and $\hat{\delta}_j$ the $i$-th and $j$-th entries of the random variable $\hat{\delta}$. For an input $x$ of dimension $d$, our smoothed classifier is defined as follows

$$g(x) \coloneqq \arg\max_y \mathbb{P}\left(f(x + \delta) = y\right) \quad \text{with } \varepsilon \sim \mathcal{N}\left(0, \frac{\sigma^2}{\sqrt[d]{\det(\Sigma)}}\Sigma\right)[1] \qquad (3)$$

We refer to this algorithm as **D**ynamic **Smooth**hing (DSMOOTH). This algorithm dynamically adapts to adversarial attacks, since the matrix $\Sigma$ embeds information on the adversarial perturbations $\hat{\delta}$. In equation 3, $\sigma$ is a user-defined parameter of the smoothed classifier. As in the original work by (Cohen et al., 2019b), the parameter $\sigma$ regulates the trade-off between robustness and accuracy. In fact adding more noise (a higher $\sigma$) tends to increase the robustness of the model to adversarial attacks, as the model's predictions become more invariant to small perturbations in the input. However, this can also degrade the model's accuracy on clean, unperturbed inputs because the predictions become more uncertain. In App. F we show examples of CIFAR-10 (Fig. 6) and IMAGENET (Fig. 7) images corrupted with the smoothing distribution as in equation 3 for a SQUARE + FGSM attack as described in Sec. 3.1. We remark that DSMOOTH as in equation 3 is essentially a generalization of the framework by Cohen et al. (2019a). In fact, by setting $\Sigma = I$ in equation 3, DSMOOTH is equivalent to the randomized smoothing algorithm in equation 1 of Cohen et al. (2019a).

**Practical implementation of the smoothing algorithm as in equation 3.** In general, the matrix $\Sigma$ in equation 3 is unknown and it has to be learned from samples. We approximate $\Sigma$ is to gather sample perturbations $\hat{\delta}$ in simulation, and then compute the resulting sample covariance matrix as in equation 2. However, the resulting smoothing algorithm as in equation 3 may be impractical when dealing with large input, since the size of $\Sigma$ grows with the input size.

To overcome this problem, we use Principal Component Analysis (PCA) (Abdi & Williams, 2010) to provide a surrogate $\Sigma_k$ of reduced size for the covariance matrix $\Sigma$, and sample $\varepsilon$ as in equation 3 using $\Sigma_k$. To generate $\Sigma_k$, we use a rank-$k$ approximation, where $k < \dim(\Sigma)$. This is done by retaining only the top $k$ eigenvectors corresponding to the largest $k$ eigenvalues. The approximated covariance matrix $\Sigma_k$ can then be expressed as $\Sigma_k = U_k \Lambda_k U_k^T$, where $U_k$ is a matrix of size $d \times k$ containing the top $k$ eigenvectors, and $\Lambda_k$ is a diagonal matrix of size $k \times k$ containing the top $k$ eigenvalues. In Sec. 5 we show empirically that different choices for $k$ do not significantly affect the performance of DSMOOTH.

Algorithms for the evaluation and certification of $g$ as in equation 3 are given in App. A.

### 4.2 CERTIFICATION GUARANTEES

We derive certification guarantees for a smoothed classifier as in equation 3. In this section, we provide guarantees based on the Mahalanobis distance, which can be seen as a generalization of the $\ell_2$ norm. However, we derive guarantees for our method in terms of the $\ell_2$ norm in Sec. 4.3. Formally, we consider the following distance in our analysis.

**Definition 4.1** (Mahalanobis Distance). *Consider adversarial examples of the form $\hat{x} = x + \hat{\delta}$ as defined in Sec. 3.1, and denote with $\Sigma$ be the covariance matrix of the r.v. $\hat{\delta}$. Then, the Mahalanobis distance of $\hat{x}$ with respect to (w.r.t.) $x$ is defined as*

$$\text{MAHL}(\hat{x} \mid x) \coloneqq \sqrt{(\hat{x} - x)^T \Sigma^{-1} (\hat{x} - x)},$$

*where $\Sigma^{-1}$ is the inverse of $\Sigma$.*

In contrast to the standard $\ell_p$ norms, the Mahalanobis distance in Def. 4.1 adjusts for the spread and orientation of the adversarial perturbations $\delta$. Since the eigenvalues of $\Sigma$ are proportional to the amount of variance captured by each principal component, then any safety boundary of the

---

[1]Throughout this work we assume that $\det(\Sigma) \neq 0$, i.e., we assume that $\Sigma$ is positive-definite.

form $\text{MAHL}(\hat{x} \mid x) \leq R$ is an ellipsoidal "stretched" in the direction of the worst-case adversarial examples. In Sec. 5 we show experimentally that certified radii based on the Mahalanobis distance are better suited for complex adversarial attacks than certification bounds based on $\ell_p$ norm. There is a natural connection between the Mahalnobis and the $\ell_2$ norm, as discussed in Sec. 4.3.

**A general framework for the push-forward measure.** Before discussing these results, however, we prove a general theoretical result, which is essential to provide guarantees for our proposed certification method. This result, which could be of independent interest, ensures general certification guarantees when the smoothing distribution is the push-forward distribution of an isotopic Gaussian distribution. Recall that the push-forward measure is defined as follows.

**Definition 4.2** (Push-forward measure). *Given a measurable space $(X, \mathcal{A})$ and a measurable function $p : X \to Y$ mapping from $X$ to $Y$, and a measure $\mu$ on $X$, the push-forward measure $p^\sharp \mu$ on $Y$ is defined as $(p^\sharp \mu)(B) = \mu(p^{-1}(B))$ for any measurable set $B \subseteq Y$.*

In other words, sampling from the push-forward measure $p^\sharp \mu$ consists of first sampling from $\mu$, and then applying the function $p$ to this sample. In the following theorem, we extend guarantees for randomized smoothing to a generic sampling distribution of the form $p^\sharp \mathcal{N}(0, \sigma^2 I)$. Throughout this section, we denote with $\Phi^{-1}$ the inverse of the standard Gaussian CDF. The following theorem holds.

**Theorem 4.3** (Randomized smoothing for the push-forward measure). *Consider a classifier $f$, and let $p$ be a deterministic invertible function. Consider the mapping $g(x) := \arg\max_y \mathbb{P}(f(x + \delta) = y)$ with $\delta \sim p^\sharp \mathcal{N}(0, \sigma^2 I)$. For a class $y_A \in Y$ suppose that there exist two constants $\underline{p_A}, \overline{p_B} \in [0, 1]$ such that*

$$\mathbb{P}(f(x + \varepsilon) = y_A) \geq \underline{p_A} \geq \overline{p_B} \geq \max_{y_B \neq y_A} \mathbb{P}(f(x + \varepsilon) = y_B),$$

*with $\varepsilon \sim p^\sharp \mathcal{N}(0, \sigma^2 I)$. Then, it holds $g(x + \delta) = y_A$ for all $\delta$ such that*

$$\left\| p^{-1}(\delta) \right\|_2 \leq \frac{\sigma}{2} \left( \Phi^{-1}(\underline{p_A}) - \Phi^{-1}(\overline{p_B}) \right).$$

Thm. 4.3 provides a defensive method that ensures guarantees in terms of the $\ell_2$ norm with respect to $p^{-1}(\delta)$. Similarly to the original safety bounds for randomized smoothing proposed by Cohen et al. (2019a), Thm. 4.3 does not require specific assumptions on the inner workings of $f$, nor the knowledge of its Lipschitz constant. Note that Thm. 4.3 is very general, since smoothing distributions such as $p^\sharp \mathcal{N}(0, \sigma^2 I)$, for different choices of $p$, allow one to sample the noise from a broad class of distributions. By choosing $p$ wisely, we can sample from smoothing distributions that are appropriate for white-box multi-attacks as in Def. 3.1. Suitable choices of $p$ may depend on the specific adversarial attacks considered. The proof of Thm. 4.4 provides an explicit choice of $p$, which is suitable for our case. Note that it is unclear what the relationship between the quantity $\left\| p^{-1}(\delta) \right\|_2$ and any known metric on the sample space $x$ is, for a generic $p$. However, we show in the remainder of this section that it is possible to derive bounds for the Mahalanobis distance and the $\ell_2$ norm using Thm. 4.3, for specific choices of $p$.

**Probabilistic guarantees for the Mahalanobis distance.** We first prove the following technical result, which allows us to build a suitable function $p$ to apply Thm. 4.3 to our case.

**Theorem 4.4.** *Consider two random variables $X \sim \mathcal{N}(x; 0, \sigma^2 I)$ and $Y \sim \mathcal{N}(y; \mu, \Sigma)$. Suppose that $\Sigma$ and $I$ have the same dimensions. Furthermore, suppose that $det(\sigma^2 I) = det(\Sigma)$. Then, there exists a deterministic invertible function $p$ such that:*

*1. $Y = p^\sharp X$;*

*2. $\sqrt{(y - \mu)^T \Sigma^{-1} (y - \mu)} = \frac{1}{\sigma} \left\| p^{-1}(y) \right\|_2$ for all $y$ in the support of $Y$.*

*Here, the function $p$ is explicitly defined as $p(x) := \frac{1}{\sigma} L x + \mu$, where $L$ be a lower-triangular matrix that gives the Cholesky decomposition of $\Sigma$.*

The proof of this theorem is deferred to App. C. By Thm. 4.4, we can apply Thm. 4.3 to the smoothing algorithm as in equation 3, to derive guarantees in terms of the Mahalanobis distance. The following lemma holds.

**Lemma 4.5** (Probabilistic Guarantees for the Mahalanobis distance). *Consider a classifier $f$, and let $g(x)$ be the corresponding smoothed classifier as in equation 3. For a class $y_A \in Y$ suppose that*

*there exist two constants $\underline{p_A}, \overline{p_B} \in [0, 1]$ such that*

$$\mathbb{P}\left(f(x + \varepsilon) = y_A\right) \geq \underline{p_A} \geq \overline{p_B} \geq \max_{y_B \neq y_A} \mathbb{P}\left(f(x + \varepsilon) = y_B\right),$$

*with $\varepsilon \sim \mathcal{N}\left(0, \frac{\sigma^2}{\sqrt[d]{det(\Sigma)}}\Sigma\right)$. Then, it holds $g(\hat{x}) = y_A$ for all adversarial samples $\hat{x}$ such that*

$$\text{MAHL}(\hat{x} \mid x) \leq \frac{\sigma}{2 \sqrt[2d]{det(\Sigma)}} \left(\Phi^{-1}(\underline{p_A}) - \Phi^{-1}(\overline{p_B})\right).$$

This lemma allows one to derive probabilistic guarantees for randomized smoothing, in terms of the distance as in Def. 4.1. The proof of this result is deferred to App. C.

### 4.3 RELATIONSHIP WITH THE $\ell_2$ NORM

In this section, we derive probabilistic guarantees for DSMOOTH, based on the $\ell_2$ norm. There is a straightforward connection between the Mahalanobis distance and the $\ell_2$ norm, as follows. For a matrix $\Sigma$ as in Def. 4.1, denote with $W$ any matrix such that $\Sigma = WW^T$. Then, it holds $\Sigma^{-1} = (W^{-1})^T W^{-1}$. Hence,

$$\text{MAHL}(\hat{x} \mid x) = \sqrt{(\hat{x} - x)^T (W^{-1})^T W^{-1}(\hat{x} - x)} = \left\| W^{-1}(\hat{x} - x) \right\|_2. \quad (4)$$

By equation 4, the Mahalanobis distance is the $\ell_2$ norm after a *whitening transformation* (Kessy et al., 2018), i.e., a linear transformation that transforms a vector of random variables $\hat{x} - x$ with a known covariance matrix $\Sigma$ into a set of new variables whose covariance is the identity matrix. In general, the matrix $W$ in equation 4 is not uniquely defined. However, the resulting $\ell_2$ norm $\left\| W^{-1}(\hat{x} - x) \right\|_2$ is equivalent across all these transformations, although some forms of $W$ may have practical advantages over others (see, e.g., (Kessy et al., 2018)). We discuss common choices of $W$ in App. D. By combining equation 4 with Lemma 4.5, we derive probabilistic guarantees for the $\ell_2$ norm as follows.

**Corollary 4.6** (Probabilistic guarantees for the $\ell_2$ norm). *Consider a classifier $f$, and $g(x)$ be the corresponding smoothed classifier as in equation 3. For a class $y_A \in Y$ suppose that there exist two constants $\underline{p_A}, \overline{p_B} \in [0, 1]$ such that*

$$\mathbb{P}\left(f(x + \varepsilon) = y_A\right) \geq \underline{p_A} \geq \overline{p_B} \geq \max_{y_B \neq y_A} \mathbb{P}\left(f(x + \varepsilon) = y_B\right),$$

*with $\varepsilon \sim \mathcal{N}\left(0, \frac{\sigma^2}{\sqrt[d]{det(\Sigma)}}\Sigma\right)$. Then, it holds $g(\hat{x}) = y_A$ for all adversarial samples $\hat{x}$ such that*

$$\left\| W^{-1}(\hat{x} - x) \right\|_2 \leq \frac{\sigma}{2 \sqrt[2d]{det(\Sigma)}} \left(\Phi^{-1}(\underline{p_A}) - \Phi^{-1}(\overline{p_B})\right),$$

*where $W$ is any matrix such that $\Sigma = WW^T$.*

We remark that, in general, the properties of $W$ in Cor. 4.5 depend on the specific covariance matrix $\Sigma$. However, if the perturbations $\varepsilon$ are sampled from an isotopic Gaussian distribution as in Cohen et al. (2019a), i.e., $\varepsilon \sim \mathcal{N}(0, \sigma^2 I)$, then Cor. 4.5 gives the same approximation guarantees as in Cohen et al. (2019a). In fact, consider a DSMOOTH algorithm with $\Sigma = \sigma^2 I$. For this algorithm, we can choose $W^{-1} = \frac{1}{\sigma}I$, and have that

$$\sqrt[d]{\det(\Sigma)} = \sqrt[d]{\det(\sigma^2 I)} = \sigma^2 \quad \text{and} \quad \left\| W^{-1}(\hat{x} - x) \right\|_2 = \frac{1}{\sigma} \left\| \hat{x} - x \right\|_2. \quad (5)$$

By substituting equation 5 in Cor. 4.5 we derive the same approximation guarantees as in Theorem 1 by Cohen et al. (2019a), which we restate for convenience.

**Corollary 4.7** (Probabilistic guarantees for isotopic Gaussian noise, equivalent to Theorem 1 by Cohen et al. (2019a)). *Consider a classifier $f$, and $g(x)$ be the corresponding smoothed classifier as in equation 3, with $\Sigma = \sigma^2 I$. For a class $y_A \in Y$ suppose that there exist two constants $\underline{p_A}, \overline{p_B} \in [0, 1]$ such that*

$$\mathbb{P}\left(f(x + \varepsilon) = y_A\right) \geq \underline{p_A} \geq \overline{p_B} \geq \max_{y_B \neq y_A} \mathbb{P}\left(f(x + \varepsilon) = y_B\right),$$

*with $\varepsilon \sim \mathcal{N}\left(0, \sigma^2 I\right)$. Then, it holds $g(\hat{x}) = y_A$ for all adversarial samples $\hat{x}$ such that*

$$\left\| \hat{x} - x \right\|_2 \leq \frac{\sigma}{2} \left(\Phi^{-1}(\underline{p_A}) - \Phi^{-1}(\overline{p_B})\right).$$

We remark that the bounds of Cor. 4.7 are known to be tight for isotopic Gaussian noise (Cohen et al., 2019a).

Table 1: Base classifiers and execution time of DSMOOTH on CIFAR-10. In this table, **Params.** (in millions) denotes the number of parameters, and **Time** (in seconds) denotes the average execution time required for a model to certify a datapoint, using the DCERT algorithm as in App. A. The execution time of DSMOOTH is similar to that of RANDSMOOTH and LSMOOTH on these base classifiers (see Table 3-5 in App. E.1).

| Model | Params. (m) | Time (s) | Model | Params. (m) | Time (s) |
|---|---|---|---|---|---|
| resnet20 | 0.27 | $4.35 \pm 0.08$ | mobilenetv2_x0_5 | 0.7 | $7.63 \pm 0.33$ |
| resnet32 | 0.47 | $5.87 \pm 0.24$ | mobilenetv2_x1_4 | 4.33 | $17.58 \pm 0.54$ |
| resnet44 | 0.66 | $6.81 \pm 0.09$ | shufflenetv2_x1_0 | 1.26 | $6.89 \pm 0.09$ |
| resnet56 | 0.86 | $7.9 \pm 0.05$ | shufflenetv2_x0_5 | 0.35 | $4.47 \pm 0.1$ |
| vgg13_bn | 9.94 | $7.43 \pm 0.1$ | shufflenetv2_x2_0 | 5.37 | $13.73 \pm 0.49$ |
| vgg16_bn | 15.25 | $10.46 \pm 0.21$ | repvgg_a0 | 7.84 | $18.74 \pm 0.72$ |
| vgg19_bn | 20.57 | $11.01 \pm 0.09$ | repvgg_a1 | 12.82 | $26.28 \pm 0.14$ |
| mobilenetv2_x1_0 | 2.24 | $12.2 \pm 0.32$ | repvgg_a2 | 26.82 | $27.65 \pm 0.26$ |

Table 2: Base classifiers and execution time of DSMOOTH on IMAGENET. In this table, **Params.** (in millions) denotes the number of parameters, and **Time** (in seconds) denotes the average execution time required for a model to certify a datapoint, using the DCERT algorithm as in App. A. The execution time of DSMOOTH is similar to that of RANDSMOOTH and LSMOOTH on these base classifiers (see Table 4-6 in App. E.1).

| Model | Params. (m) | Time (s) | Model | Params. (m) | Time (s) |
|---|---|---|---|---|---|
| resnet50 | 25.56 | $26.73 \pm 0.21$ | wide_resnet50_2 | 68.88 | $40.24 \pm 0.78$ |
| resnet152 | 60.19 | $107.52 \pm 8.04$ | wide_resnet101_2 | 126.89 | $86.91 \pm 11.97$ |

## 5 EXPERIMENTS

The overall aim of the experiments is to demonstrate that DSMOOTH achieves good *certified accuracy* compared to baselines on complex adversarial attacks as in Def. 3.1. The certified accuracy is defined as the fraction of the test set, which a smoothed algorithm classifies correctly with a prediction that is certifiably robust within a ball of a given radius. Since DSMOOTH is a randomized smoothing classifier, it is not possible to compute this quantity exactly. Instead, we report on the approximate certified test set accuracy following previous related work, e.g., Cohen et al. (2019a). In addition to evaluating the certified accuracy, we also report on the execution time of DSMOOTH, and its sensitivity to different choices of the parameter $k$ for the $k$-rank approximation as in Sec. 4.1.

In all the experiments we consider the SQUARE + FGSM multi-attack as detailed in Sec. 4.1, obtained as a combination of the Square Attack algorithm (Andriushchenko et al., 2020) and FGSM (Goodfellow et al., 2015). This attack applies a Square Attack to an input image (using $\ell_\infty$ norm), and then it applies a FGSM attack to the resulting adversarial sample (using $\ell_2$ norm). In this experiment we use Square Attack with maximum perturbation 0.5 and 5000 queries. The FGSM attack uses maximum perturbation parameter 0.5. In App. F we show examples of CIFAR-10 (Fig. 6) and IMAGENET (Fig. 7) images corrupted with the smoothing distribution as in equation 3 for this type of attack.

### 5.1 OVERALL SET-UP

**Base classifiers training.** We consider various pre-trained classifiers, that achieve high accuracy on CIFAR-10 and IMAGENET respectively (see Table 1-2). We then fine-tune these classifiers to improve the robustness to adversarial attacks to SQUARE + FGSM as detailed in Sec. 3.1. Pre-trained models on CIFAR-10 (Table 1) are downloaded from https://github.com/chenyaofo/pytorch-cifar-models, and pre-trained models on IMAGENET (Table 2) are downloaded from https://github.com/pytorch/pytorch. Fine-tuning consists of adjusting these models to a dataset that contains both CIFAR-10 training images and adversarial examples. The ratio of natural and adversarial examples is $50 : 50$. In this work, we opt for a simple training procedure, to highlight the benefits of our method against baselines. However, we believe that the certified accuracy of our method could be further improved by considering more complex adversarial training procedures, such as Wong & Kolter (2021).

**Baselines.** We compared our smoothing algorithm in equation 3 to two baseline approaches for certified robustness: the standard randomized smoothing algorithm by (Cohen et al., 2019a) (RANDSMOOTH), and the approach by Teng et al. (2020) (LSMOOTH). The randomized smoothing algorithm by Cohen et al. (2019a) provides certification guarantees in terms of the $\ell_2$ norm, whereas the algorithm by Teng et al. (2020) provides guarantees in terms of the $\ell_1$ norm. We do not consider randomized smoothing techniques with certification guarantees in terms of $\ell_p$ norms with $p > 2$, since impossibility results are known for increasing $p$ (Yang et al., 2020; Blum et al., 2020; Kumar et al., 2020). Specifically, we do not consider any certification mechanism for the $\ell_\infty$ norm, since the isotopic Gaussian distribution as in RANDSMOOTH is optimal for defending against $\ell_\infty$ attacks, if we don't use a more powerful technique than Neyman-Pearson (Yang et al., 2020).

**System.** The system used features multiple Intel® Xeon® Gold 6252 CPUs, each with a base clock speed of 2.10 GHz, operating at various frequencies between 2011 MHz and 2800 MHz. The system also includes six NVIDIA GPUs for more intensive graphics and computational workloads. These are two NVIDIA GPUs with a 64-bit width and clock speed of 33 MHz, and four NVIDIA GPUs of the GV102 model with a 64-bit width, operating at a clock speed of 33 MHz.

## 5.2 RESULTS ON CIFAR-10

**Execution time.** We test the performance of DSMOOTH. To this end, we run the DCERT algorithm, as detailed in App. A, on various base models. Parameters for DCERT are $\alpha = 0.001$, $n_0 = 100$ Monte Carlo samples for selection and $n = 100000$ samples for estimation. With this parameters choice, there is at most $0.001$ probability that DCERT returns a radius that is not robust (see App. A). For each base classifier, we test our algorithm on $500$ images from CIFAR-10 and we report on the average execution time (in seconds) to certify a single image. The results are reported in Table 1. Overall we observe that DSMOOTH is scalable to all models, although for models with several million parameters, such as RepVGG_a2, the performance decreases. We remark that the performance of our algorithm is similar to the performance of previous algorithms, e.g., the algorithms by Cohen et al. (2019a); Teng et al. (2020). We refer the reader to Table 3-5 in App. E.1 for the execution time of previous algorithms.

**Comparison against baselines.** We run the DCERT algorithm (App. A) against baselines with parameters $\alpha = 0.001$, $n_0 = 100$ samples for selection and $n = 100000$ samples for estimation. For each base classifier in Table 1, we test DCERT and baselines on $500$ images from CIFAR-10. The results are displayed in Fig. 1, where we observe that in all cases our algorithm performs significantly better than the baselines. These results demonstrate that DSMOOTH is suitable to handle complex adversarial attacks as in Def. 3.1, whereas RANDSMOOTH and LSMOOTH are unsuitable to that end. In fact, in most cases the certified accuracy of RANDSMOOTH and LSMOOTH is approximately $0.1$. Since CIFAR-10 has only 10 classes, these results suggest that RANDSMOOTH and LSMOOTH do not perform significantly better than uniform random sampling.

**Additional experiments.** In App. E.2 we provide additional experiments on CIFAR-10 to determine the effect of different choices of $\alpha$ and number of samples for selection $n$ on the performance of DCERT.

## 5.3 RESULTS ON IMAGENET

**Execution time.** We evaluate the effectiveness of our smoothing algorithm as described in equation 3. To achieve this, we apply the DCERT algorithm, as outlined in App. A, across different base models. For DCERT, we use parameters $\alpha = 0.001$, $n_0 = 100$ Monte Carlo samples for selection, and $n = 1000$ samples for estimation. In this scenario, we approximate the matrix $\Sigma$ from equation 2 using a PCA algorithm, as explained in Sec. 4.1, with a rank-$k$ approximation where $k = 1000$. Each base classifier is tested on $500$ images from IMAGENET, and we measure the average execution time (in seconds) required to certify a single image. The results are summarized in Table 2. Overall, DSMOOTH demonstrates scalability to very large models, and its performance is comparable to that of previous algorithms (see Table 3-5 in App. E.1).

**Comparison against baselines.** Once again, we evaluated our smoothing algorithm from equation 3 against baselines. We apply the DCERT algorithm (see App. A) with parameters $\alpha = 0.001$, $n_0 = 100$ samples for selection, and $n = 1000$ samples for estimation. Due to the large size of $\Sigma$, we use a rank-$k$ approximation $\Sigma_k$ with $k = 1000$. For each base classifier listed in Table 2, we evaluate DCERT and the baseline methods on $500$ images from CIFAR-10. The results are presented in Fig. 2,

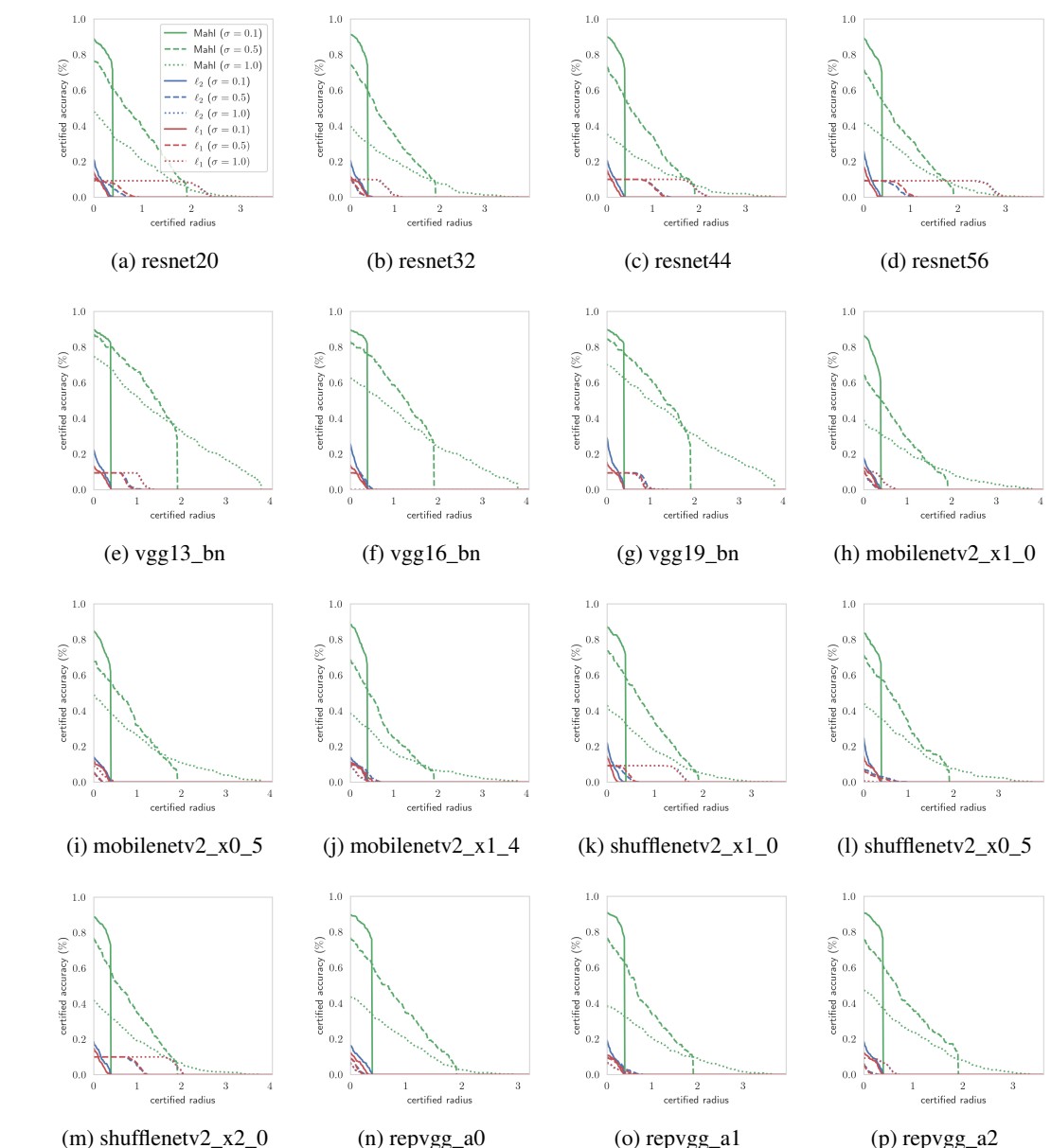

Figure 1: Approximate certified accuracy of DSMOOTH (MAHL in the legend), RANDSMOOTH ($\ell_2$ in the legend) and LSMOOTH ($\ell_1$ in the legend) on CIFAR-10 for various base models as in Table 1. DSMOOTH is significantly better than baselines.

showing that our algorithm consistently outperforms the baselines. As with the CIFAR-10 results, this demonstrates that DSMOOTH is effective against complex adversarial attacks as defined in Def. 3.1, while RANDSMOOTH and LSMOOTH are inadequate for this purpose.

**Ablation study on the rank-$k$ approximation of $\Sigma$.**   We conclude with a set of experiments to determine if our results are sensitive to the rank $k$ of the PCA approximation of the covariance matrix $\Sigma$. To this end, we run the DCERT algorithm with the smoothing distribution as in equation 3, for $k = 10, 100, 1000, 10000$. Each run uses the parameters $\sigma = 0.5$, $\alpha = 0.001$, $n_0 = 100$ samples for selection and $n = 1000$ samples for estimation. The results are displayed in Fig. 3. The results suggest that DSMOOTH is not very sensitive to different choices of $k$.

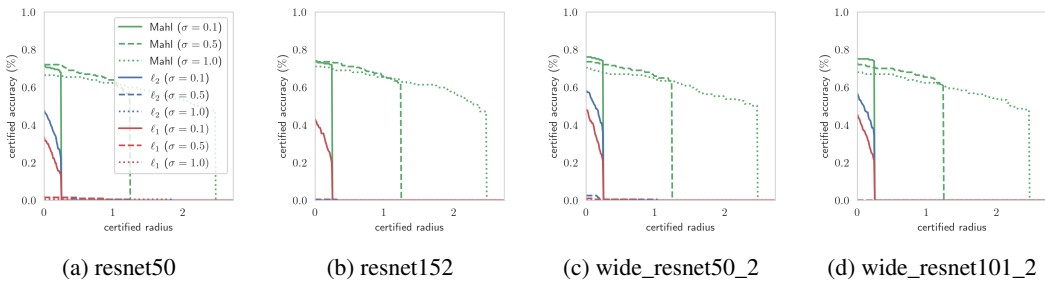

|     |     |     |     |
| :-: | :-: | :-: | :-: |
| (a) resnet50 | (b) resnet152 | (c) wide_resnet50_2 | (d) wide_resnet101_2 |

Figure 2: Approximate certified accuracy of DSMOOTH (MAHL in the legend), RANDSMOOTH ($\ell_2$ in the legend) and LSMOOTH ($\ell_1$ in the legend) on IMAGENET for various base models as in Table 2. We observe that DSMOOTH comes out on top.

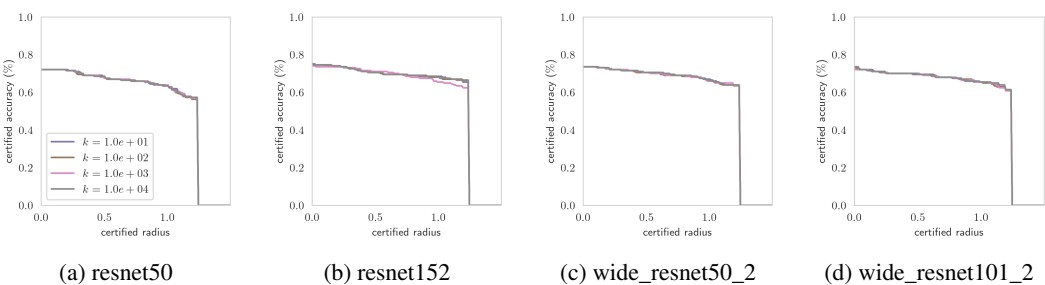

|     |     |     |     |
| :-: | :-: | :-: | :-: |
| (a) resnet50 | (b) resnet152 | (c) wide_resnet50_2 | (d) wide_resnet101_2 |

Figure 3: Approximate certified accuracy of DSMOOTH for different choices of $k$ on IMAGENET, for various base models as in Table 2. We observe that the parameter $k$ does not significantly affect the performance of DSMOOTH.

## 6 DISCUSSION

In this paper, we introduced a novel certification method based on randomized smoothing (see equation 3) to enhance the robustness of machine learning models against complex adversarial attacks, including combinations of multiple attack types (see Sec. 4.1). Our approach generalizes the existing framework of randomized smoothing by incorporating more flexible noise distributions, allowing for robustness guarantees across a wider range of adversarial threats, such as SQUARE+FGSM (see Sec. 4.1). Through extensive experiments on CIFAR-10 (see Sec. 5.2) and IMAGENET (see Sec. 5.3), we demonstrated that our method consistently outperforms state-of-the-art defenses in terms of certified accuracy (see Fig. 1-2) .

However, much like previous work (see, e.g., Cohen et al. (2019a); Teng et al. (2020)), our proposed method still faces several limitations. The effectiveness of DSMOOTH is constrained by its reliance on Monte Carlo sampling, which can be computationally expensive on very large models. Additionally, while our approach extends robustness beyond the standard $\ell_2$ norm, it may not yet fully capture the complexities of all possible adversarial threats.

Future work could address these limitations by developing more efficient sampling techniques, or by leveraging neural architecture search to identify base classifiers that are inherently more robust to adversarial perturbations. Furthermore, exploring alternative noise distributions and adaptive smoothing strategies could further enhance robustness against a broader array of adversarial threats.

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
