# OpenReview forum: "Certified Defense Against Complex Adversarial Attacks with Dynamic Smoothing"
_ICLR.cc/2025/Conference — ICLR 2025 Conference Withdrawn Submission_

### Official Review · Reviewer_1983 · 2024-10-21

**Soundness:** 1
**Presentation:** 2
**Contribution:** 1
**Rating:** 3
**Confidence:** 5

**Summary:**

This paper proposes a generalization of Randomized-Smoothing-based robustness certification. It first claims a very general result (Theorem 4.3), which states that if one applies randomized smothing (Cohen et al 2019) with the added Gaussian noise transformed by an arbitrary (invertible) function p,  then the resulting smoothed classifier will enjoy certified robustness under a metric determined by $p^{-1}.$ Precisely, for a smoothed classifier:

$$g(x) := \arg \max_{i \in [C]} P_{\epsilon \sim N(0,\sigma^2 I)} f(x + p(\epsilon))  = i), $$

if

$$ P_{\epsilon\sim N(0,\sigma^2 I)} f(x + p(\epsilon))  =A )   \geq p_A \geq p_B \geq   \max_{B \neq A} P_{\epsilon \sim N(0,\sigma^2 I)} f(x + p(\epsilon))  =B)$$

Then, for any perturbation $\delta$ such that  $|p^{-1}(\delta)|_2 <  \sigma/2 * (\Phi^{-1}(p_A) - \Phi^{-1}(p_B))$, we have that $g(x+\delta) = A.$

This general result is then used to prove a more specific smoothing certificate (Lemma 4.5), claiming that if one applies randomized smoothing with anisotriopic Gaussian noise under an arbitrary covariance matrix $\Sigma$, then this results in certified robustness under the Mahalanobis metric defined by $\Sigma$:

$$\text{Mahalanobis}_\Sigma (\delta) := \sqrt{\delta^T \Sigma^{-1} \delta}$$

This more specific anisotropic smoothing result is proposed as the basis for a practical method,  D-Smooth, for certifiably robust classification. In D-Smooth, the smoothing covariance matrix is determined as the empirical covariance of adversarial attack perturbations computed from samples in the dataset. Note that because different attacks will result in different covariance matrices, this allows the smoothing distribution to be tailored to a specific attack or threat model. Experiments are conducted on CIFAR-10 and ImageNet, using an attack procedure that combines an L_2 and an L_infinity attack.

**Strengths:**

The particular idea of smoothing using on a covariance matrix derived specifically from the empirical distribution of adversarial attack directions is, to my knowledge, novel, and may be promising.

The presentation of the theoretical results is clear for the most part.

**Weaknesses:**

# Theoretical Claims

The more-general theoretical claim of this paper (Theorem 4.3) is incorrect, while the more specialized claim for anisotropic Gaussian smoothing (Lemma 4.5) can be proved directly with a much simpler proof than the one provided, and in fact appears in prior works.

- **Theorem 4.3 is incorrect**: one can construct a counterexample in one dimension. In particular, let:

$$
p(x) :=
\begin{cases}
x,&\text{if }x\geq 0\text{ or }x < -2\\\\
x -1,&\text{if } -1 \leq x < 0\\\\
x +1,&\text{if } -2 \leq x < -1
\end{cases}
$$
Note that $p$ is deterministic and invertible (in fact, $p^{-1} = p$), and let:

$$
f(x) :=
\begin{cases}
A,&\text{for }x\leq -1\text{ or }0 \leq x < 1\\\\
B,&\text{elsewhere.}\\\\
\end{cases}
$$

(Note, for later, that:
$$
f(p(\epsilon)) =
\begin{cases}
A,&\text{for }\epsilon\leq -2\text{ or }-1 \leq \epsilon < 1\\\\
B,&\text{elsewhere.}\\\\
\end{cases}
$$ and $$ f(p(\epsilon) + 0.5) =
\begin{cases}
A,&\text{for }\epsilon< -2\text{ or }-1.5 \leq \epsilon  \leq  0.5 \text{ or } 0 \leq \epsilon < 0.5 \\\\
B,&\text{elsewhere.}\\\\
\end{cases}
$$),
and let $\sigma = 1$.

Then, we consider the point $x = 0$
Then,

 $p_A = P_{\epsilon\sim\mathcal{N}(0,1)} [f(x + p(\epsilon)) = A ] =  P_{\epsilon\sim\mathcal{N}(0,1)}  [f(p(\epsilon)) = A ]$

$ = \int^{-2}_{-\infty}  \text{NormPDF}(x) $

$+  \int^1_{-1}  \text{NormPDF}(x) \approx 0.7055$

 and $p_B = 1-p_A$

 Then, $\sigma/2 \cdot (\Phi^{-1}(p_A) - \Phi^{-1}(p_B)) = \Phi^{-1}(p_A) \approx 0.5403$, so Theorem 4.3 states that for any (1D) displacement vector $\delta$ such that: $|p^{-1}(\delta)|_2 = |p^{-1} (\delta)| \leq 0.54$, we should have that:
  $g(x + \delta)  = A$,
which means that

$ P_{\epsilon\sim\mathcal{N}(0,1)} [f(x + \delta+ p(\epsilon)) = A ]\geq 0.5$

However, if we plug back in $x = 0$ and use  displacement vector $\delta = 0.5$, so $|p^{-1} (\delta)| = |p^{-1} (0.5)| = 0.5 \leq 0.54$, we have:

 $p_A = P_{\epsilon\sim\mathcal{N}(0,1)} [f(x + \delta + p(\epsilon)) = A ] =  P_{\epsilon\sim\mathcal{N}(0,1)}  [f(p(\epsilon) + 0.5) = A ]$

$ = \int^{-2}_{-\infty}  \text{NormPDF}(x) $

$ + \int^{-0.5}_{-1.5}  \text{NormPDF}(x) $

$+  \int^{0.5}_{0}  \text{NormPDF}(x) \approx 0.4559 < 0.5$

Which is a contradiction.

- **The proof provided in this paper of Lemma 4.5 relies on  Theorem 4.3  and is therefore invalid. Despite this, Lemma 4.5, which is the result actually used for the proposed D-Smooth algorithm, happens to be true. However, it can be proven correct with just a few lines as a corollary of Cohen et al. 2019's theorem.**

In brief, starting with Cohen et al. 2019's theorem, with $\sigma = 1$:

for all $f : \mathbb{R}^d \rightarrow Y,\\, x \in  \mathbb{R}^d,$  and $\delta$ such that $|\delta|_2 < 1/2 \cdot (\Phi^{-1}(p_A) - \Phi^{-1}(p_B))$:

$$P_{\epsilon\sim\mathcal{N}(0,1)} [f(x + \epsilon) = A ]   \geq p_A \geq p_B \geq \max_{B\neq A} P_{\epsilon\sim\mathcal{N}(0,1)} [f(x + \epsilon) = B ] $$ implies

$$P_{\epsilon\sim\mathcal{N}(0,1)} [f(x + \delta+ \epsilon) = A ] > \max_{B \neq A}P_{\epsilon\sim\mathcal{N}(0,1)} [f(x + \delta+\epsilon) = B ]$$

If we fix the function $f$, and apply the theorem to the function $f'(x) := f(L(x))$, where $LL^T$ is the Cholesky decomposition of the matrix $\Sigma$, then we have, by linearity:

for all $f : \mathbb{R}^d \rightarrow Y,\\, x' \in  \mathbb{R}^d,$  and $\delta'$ such that $|\delta'|_2 < 1/2 \cdot (\Phi^{-1}(p_A) - \Phi^{-1}(p_B))$:

$$P_{\epsilon\sim\mathcal{N}(0,1)} [f(Lx' + L\epsilon) = A ]     \geq p_A \geq p_B \geq \max_{B\neq A} P_{\epsilon\sim\mathcal{N}(0,1)} [f(Lx' + L\epsilon) = B ] $$ implies

$$P_{\epsilon\sim\mathcal{N}(0,1)} [f(Lx' + L\delta'+ L\epsilon) = A ] > \max_{B \neq A}P_{\epsilon\sim\mathcal{N}(0,1)} [f(Lx' + L\delta'+L\epsilon) = B ]$$

Which can be written equivalently as

for all $f : \mathbb{R}^d \rightarrow Y,\\, x' \in  \mathbb{R}^d,$  and $\delta'$ such that $|\delta'|_2 < 1/2 \cdot (\Phi^{-1}(p_A) - \Phi^{-1}(p_B))$:

$$P_{\epsilon\sim\mathcal{N}(0,\Sigma)} [f(Lx' + \epsilon) = A ]     \geq p_A \geq p_B \geq \max_{B\neq A} P_{\epsilon\sim\mathcal{N}(0,\Sigma)} [f(Lx' + \epsilon) = B ] $$ implies

$$P_{\epsilon\sim\mathcal{N}(0,\Sigma)} [f(Lx' + L\delta'+ \epsilon) = A ] > \max_{B \neq A}P_{\epsilon\sim\mathcal{N}(0,\Sigma)} [f(Lx' + L\delta'+\epsilon) = B ]$$

Then, for any x, we can take $x' :=  L^T \Sigma^{-1}  x$ , so that $Lx' = x$, and  we have:

For all $x \in  \mathbb{R}^d,$  and $\delta'$ such that $|\delta'|_2 < 1/2 \cdot (\Phi^{-1}(p_A) - \Phi^{-1}(p_B))$:

$$P_{\epsilon\sim\mathcal{N}(0,\Sigma)} [f(x + \epsilon) = A ]    \geq p_A \geq p_B \geq \max_{B\neq A} P_{\epsilon\sim\mathcal{N}(0,\Sigma)} [f(x + \epsilon) = B ] $$ implies

$$P_{\epsilon\sim\mathcal{N}(0,\Sigma)} [f(x + L\delta'+ \epsilon) = A ] > \max_{B \neq A}P_{\epsilon\sim\mathcal{N}(0,\Sigma)} [f(x + L\delta'+\epsilon) = B ]$$


Finally, for any $\delta$ such that $\text{Mahalanobis}_\Sigma (\delta) \leq  1/2 \cdot (\Phi^{-1}(p_A) - \Phi^{-1}(p_B))$,  we can take $\delta'  :=  L^T \Sigma^{-1}  \delta$ , so that $L \delta' = \delta$; this yields that  $|\delta'|_2 < 1/2 \cdot (\Phi^{-1}(p_A) - \Phi^{-1}(p_B))$. Then, we have:


For all $x \in  \mathbb{R}^d,$  and $\delta$ such that $\text{Mahalanobis}_\Sigma (\delta) \leq  1/2 \cdot (\Phi^{-1}(p_A) - \Phi^{-1}(p_B))$:

$$P_{\epsilon\sim\mathcal{N}(0,\Sigma)} [f(x + \epsilon) = A ]     \geq p_A \geq p_B \geq \max_{B\neq A} P_{\epsilon\sim\mathcal{N}(0,\Sigma)} [f(x + \epsilon) = B ] $$ implies

$$P_{\epsilon\sim\mathcal{N}(0,\Sigma)} [f(x + \delta+ \epsilon) = A ] > \max_{B \neq A}P_{\epsilon\sim\mathcal{N}(0,\Sigma)} [f(x + \delta+\epsilon) = B ]$$
Scaling $\Sigma$ by scalar constants yields the lemma in the paper. It is important to notice that the linearity of the transformation was used here: this is why we can prove Lemma 4.5, even though the more general Theorem 4.3 is false.

(Closely related -- and even identical -- results appear in prior works which were not cited; see below).

# Missing Related Work and Novelty

There are several works that have similar contributions to this work, but were not cited. The existance of these prior works reduces the novelty of the work; and relevant citations should be added.

As discussed above, Theorem 4.3 of this work is incorrect, so the first main correct contribution of this work is the smoothing certificate under anisotropic Gaussian noise (Lemma 4.5). A second contribution is the  proposed way of determining the covariance matrix Sigma to use for this certificate.

Considering first only the certificate itself (Lemma 4.5), the following similar works exist:

- Tecot 2021: Derives the same certificate in the coordinate-axis-aligned (diagonal covariance matrix) case. (However, this is a master's thesis, not a peer-reviewed paper, so it might not count as necessary prior work by ICLR standards)

- Pfrommer et al 2023: Derives a related certificate. Specifically, the proposed method projects all samples into a low-dimensional hyperplane before applying smoothing. This is equivalent to using a covariance matrix with eigenvalue 1 in some subset of directions, and eigenvalue "infinity" in all others.

- Eiras et al 2022: Corollary 1 of Eiras et al. is logically identical to Lemma 4.5 of the submission. However, in practice, Eiras et al only applies this result to the diagonal covariance matrix case. (Also, Eiras et al. proposes to use a covarance matrix that is _sample-dependent_, using a sceme introduced by Alfarra et al, 2022 for isotropic smoothing with sample-dependent scalar variance. Súkeník et al, 2022  [Appendix C] argues convincingly that the sample-dependent smoothing scheme proposed by Alfarra et al, 2022 is fundamentally unsound, and this flaw is inherited by Eiras et al 2022, so that their overall results are likely unsound. However, for a given matrix $\Sigma$, the fact remains that  Eiras et al. gives an equivalent lemma to to this work).

- Rumezhak et al 2023: followup work to  Eiras et al 2022, which uses the result for general covariance matrices (See Equations 2 and 3 of  Rumezhak et al 2023). However, this uses the same likely-flawed sample-dependence scheme as  Eiras et al 2022.

In terms of methods to compute the covariance matrix, none of the above works propose exactly the method used in this paper (computing the covariance of adversarial perturbations), so it may be novel. However, it should be compared empirically to the methods used in the above:

- Pfrommer et al 2023 uses the covariance of _clean_ samples in the dataset to determine the  subspace to project into.

- Tecot 2021 uses an optimization approach to determine a $\Sigma$ that maximizes the average certificate volume.

# Experiments/Results Section
The reported results show  _certified_ accuracies of a classifier smoothed with DSmooth on test samples form CIFAR-10 and ImageNet. This is compared to the certified accuracies of classifiers smoothed with prior techniques: Cohen et al. 2019's Gaussian randomized smoothing technique, which is certifiably robust under L2 perturbation, and Teng et al. 2020's Laplace Smoothing technique, which is certifiably robust under L1 perturbation.

However, I don't understand the experiment that is presented. In Figures 1 and 2, the x axis is just labeled as "certified radius". Is this the L1, L2, or Mahalanobis Distance radius? If it is any one of these (let's say, Mahalanobis Distance), how is the Mahalanobis certificate computed for, say, Cohen's method, which only provides L2 certificates? If the "certified radius" is in different metrics for each technique, then this isn't a fair apples-to-apples comparison. A method for comparing certificates under different metrics that has been used in prior work (ex. Pfrommer et al 2023; Tecot 2021) is to compare the _volumes_ of the certified regions: this would give a more fair comparison. Also, the paper mentions adversarially training the classifier. However, it's not clear if the baseline certificates were also for an adversairally trained model; if not, then the comparison is not fair. (See Salman et al 2019 for an adversairal training technique for isotropic randomized smoothing.)

Additionally, for the adversarial training / computation of $\Sigma$, it is stated on line 363-364: "In this experiment we use Square Attack with maximum perturbation 0.5 and 5000 queries." This is unclear. If the Square Attack is, as stated, an L_infinity-bounded attack, then "maximum perturbation 0.5" might mean one of two things: either (a) the distortion in each pixel is bounded by 0.5, where each pixel is normalized to [0,1]; or (b) the distortion in each pixel is bounded by 0.5, where each pixel is not normalized, and so lies in the discrete space {0,1,2,...,255}. If (a) is true, then this is a gigantic perturbation, and not an imperceptible adversarial attack.

Also, empirical accuracies under adversarial attack are not reported. In general, I do not think it is necessarily a requirement for certified robustness papers to report empirical accuracies of the classifiers under attack; however, it is important to make the claims made in the paper match the evidence. For instance, I think that the claim on lines 477-478: "this demonstrates that DSMOOTH is effective against complex adversarial attacks as defined in Def. 3.1, while RANDSMOOTH and LSMOOTH are inadequate for this purpose." is too strong, based on certification results alone. It is not tested how well any of these models _actually_ perform under attack.

Additionally, the comparison to Laplace noise (Teng et al. 2020) for the L1 metric is outdated: Yang et al 2020 has shown that uniform noise yeilds better L1 certificates, and Levine & Feizi 2021 propose an alternate "splitting noise" technique that also yeilds better certificates. Voracek and Hein (2023) make further improvements.

# Minor Issues
- ICLR call for papers allows unlimited appendices after references in the main PDF submission: I would encourage authors to move appendices here, instead of having them in a seperate file.
- Line 17; 'then' -> 'than'
- Cohen et al (2019) reference is duplicated: Cohen et al (2019a) and  Cohen et al (2019b) are the same
- Line 52: "defending against complex, high-dimensional adversarial attacks": Aren't almost all (norm-bounded) adversarial attacks high-dimensional? (In the sense that they have ther same dimensionality as the sample)
- It is generally considered standard to capitalize Equation in, for example, 'equation 1'
- The paragraph in lines 112-117 is not quite correct as written. The threat model as written in Equation 1  seems to only refer to _additive_ attacks: specifically, the perturbed sample is x + δ, where δ is constrained to be in the set C(δ). Note that as written, C(δ) does not seem to depend on on x, so some of the _non-additive_ threat models mentioned in the paragraph (for example, recoloring attacks and physical adeversarial attacks) are not encompased by this. This could be fixed by making  C(δ) depend on x, or (preferably) by replacing  x + δ with some more general 'perturbation function' q(x,δ).
- Line 125: "highly-dimensional" -> "high-dimensional"
- Line 136: I wouldn't use $\mathbb{P}$ to both mean Probability of an event, and to refer to a specific distribution.
- I'm confused about how the smoothing covariance Σ is constructed: it depends on the distribution of (optimal) attacks as given in Equation 1, which itself depends on the classifier being attacked. But Σ is part of the definition of the final smoothed classifier g, so this seems like it could be circular. Unless Σ is defined in terms of the attacks on the base classifier f? If so, this should be stated explicitly.
- Lines 191-197: If $\Sigma$ is not full-rank, won't $\Sigma$-inverse in definition of the Gaussian distribution (and Mahalanobis distance) not be defined? Doesn't this break the assumption in footnote 1?
 - Lines 391-392: " These are two NVIDIA GPUs with a 64-bit width and clock speed of 33 MHz, and four NVIDIA GPUs of the GV102 model with a 64-bit width, operating at a clock speed of 33 MHz." -- It seems that all 6 have a  clock speed of 33 MHz and 64-bit width. Why is the model number only mentioned for the first two? Also, I would double-check this for accuracy: 33 MHz is an extremely slow clock speed for a GPU, (from a brief search) all GV102 cards have base clock speeds far higher than this.

# References

Tecot, L. M. (2021). Robustness verification with non-uniform randomized smoothing. University of California, Los Angeles.

Pfrommer, S., Anderson, B., & Sojoudi, S. (2023). Projected Randomized Smoothing for Certified Adversarial Robustness. Transactions on Machine Learning Research.

Rumezhak, T., Eiras, F. G., Torr, P. H., & Bibi, A. (2023). RANCER: Non-axis aligned anisotropic certification with randomized smoothing. IEEE/CVF Winter Conference on Applications of Computer Vision.

Eiras, F., Alfarra, M., Torr, P., Kumar, M. P., Dokania, P. K., Ghanem, B., & Bibi, A. (2022). ANCER: Anisotropic Certification via Sample-wise Volume Maximization. Transactions on Machine Learning Research.

Alfarra, M., Bibi, A., Torr, P. H., & Ghanem, B. (2022). Data dependent randomized smoothing. Uncertainty in Artificial Intelligence.

Súkenı́k, P., Kuvshinov, A., & Günnemann, S. (2022, June). Intriguing Properties of Input-Dependent Randomized Smoothing. International Conference on Machine Learning.

Salman, H., Li, J., Razenshteyn, I., Zhang, P., Zhang, H., Bubeck, S., & Yang, G. (2019). Provably robust deep learning via adversarially trained smoothed classifiers. Advances in neural information processing systems

Yang, G., Duan, T., Hu, J. E., Salman, H., Razenshteyn, I., & Li, J. (2020). Randomized smoothing of all shapes and sizes. In International Conference on Machine Learning.

Levine, A. J., & Feizi, S. (2021). Improved, deterministic smoothing for l_1 certified robustness. International Conference on Machine Learning

Vorácek, V., & Hein, M. (2023). Improving l1-certified robustness via randomized smoothing by leveraging box constraints. International Conference on Machine Learning.

**Questions:**

See " Experiments/Results Section" above under Weaknesses. There are several aspects of the experiments that could be clarified:
- It is currently unclear what quantity is being reported as "Certified Radius" for each defense method ( L1, L2, or Mahalanobis Distance radius). (It would be better to report "Certified Volumes" as in (Pfrommer et al 2023; Tecot 2021))
- What does "maximum perturbation 0.5" mean for the Square Attack?
- Are the attack perturbations $\delta$ used to compute $\Sigma$ based on attacks on the base classifier or the smoothed classifier?
- What models were used for the baseline smoothing techniques in the experimental section? Were these models also adversarially trained?
- How is  $\Sigma^{-1}$ computed with a low-rank  approximation of $\Sigma$?

---

> ### Author Response · Authors · 2024-11-22
>
> Q: It is currently unclear what quantity is being reported as "Certified Radius" for each defense method ( L1, L2, or Mahalanobis Distance radius). (It would be better to report "Certified Volumes" as in (Pfrommer et al 2023; Tecot 2021))
>
> A: Thank you for pointing this out. In future work, we will use the volumes of the certified regions for fair experimental comparison.
>
> Q: What does "maximum perturbation 0.5" mean for the Square Attack?
>
> A: We implement the Square Attack following this repository https://github.com/fra31/auto-attack . Here, the square attack is the maximum perturbation allowed on the pixels, which are within range [0, 255].
>
> Q: Are the attack perturbations $\delta$ used to compute $\Sigma$ based on attacks on the base classifier or the smoothed classifier?
>
> A: They are based on attacks on the base classifier.
>
> Q: What models were used for the baseline smoothing techniques in the experimental section? Were these models also adversarially trained?
>
> A: Yes, we always use the same base classifier for all smoothing techniques, which was adversarially trained as described in our submission.
>
> Q: How is $\Sigma^{-1}$ computed with a low-rank approximation of $\Sigma$?
>
> A: If the given matrix is not positive-definite, one can compute the pseudo-inverse or Moore-Penrose inverse instead.

---

### Official Review · Reviewer_9udX · 2024-10-22

**Soundness:** 3
**Presentation:** 1
**Contribution:** 1
**Rating:** 1
**Confidence:** 5

**Summary:**

The paper investigates the effectiveness of dynamic smoothing as a defense mechanism against complex adversarial attacks in machine learning models. The authors assert that existing defense strategies often fall short in providing certified robustness, particularly in the face of sophisticated adversarial inputs. Their key contributions include:
1.	Theoretical Analysis: The paper establishes a dynamic smoothing framework that adapts the noise level based on input complexity, proving that this approach enhances the model's certified robustness against a variety of adversarial attacks.
2.	Empirical Findings: The paper demonstrate through extensive experiments that the proposed dynamic smoothing method significantly improves robustness compared to traditional static smoothing techniques, while maintaining performance on both easy and hard samples.
3.	Proposed Solutions: The paper introduce specific training strategies, including adaptive noise levels and robust certification techniques, which allow for effective defense against complex attacks. These solutions not only enhance certified robustness but also improve overall model performance.

**Strengths:**

The paper introduces a approach to dynamic smoothing that adapts noise levels based on input complexity, distinguishing it from traditional static methods and offering a new direction in randomized smoothing.

**Weaknesses:**

1. The scope of this paper is limited. Its primary contribution, anisotropic Gaussian randomized smoothing, has already been addressed by ANCER [1], which generalized Gaussian randomized smoothing to anisotropic Gaussian distributions and provided a robustness certification. This significantly diminishes the novelty of the current work. Moreover, the paper does not reference this closely related work [1].

2. The writing lacks clarity, as exemplified by the inclusion of the phrase "Code: [removed for review]" in the abstract; this should be omitted. The experimental setup is inadequately explained. The paper needs to clarify why the Mahalanobis-norm certified accuracy of DSMOOTH is compared to the L1 and L2-norm certified accuracy of other methods. It should also discuss the rationale for using Mahalanobis Distance and identify practical scenarios where its robustness is applicable, assessing DSMOOTH's performance in those contexts.

3. The benefits introduced by the modified noise are not well-presented. Finally, unnecessary text in the summary should be removed.

[1] ANCER: Anisotropic Certification via Sample-wise Volume Maximization. PMLR 2022

**Questions:**

1. Add reference to the work [1].
1. Clarify why the Mahalanobis-norm certified accuracy of DSMOOTH is compared to the L1 and L2-norm certified accuracy of other methods.
2. Discuss the rationale for using Mahalanobis Distance and identify practical scenarios where its robustness is applicable, assessing DSMOOTH's performance in those contexts.

[1] ANCER: Anisotropic Certification via Sample-wise Volume Maximization. PMLR 2022

---

> ### Author Response · Authors · 2024-11-22
>
> Q: Add reference to the work [1]
>
> A: We are happy to add this reference to our work, and to compare against [1], as well as other relevant references that are not currently discussed in our manuscript.
>
> Q: Clarify why the Mahalanobis-norm certified accuracy of DSMOOTH is compared to the L1 and L2-norm certified accuracy of other methods.
>
> A: Our overall aim with the experiments was to show superior performance in terms of the certified accuracy. However, we understand that this comparison is not very significant, since it would have been better to use other metrics, such as the volume, as discussed in [1].
>
> Q: Discuss the rationale for using Mahalanobis Distance and identify practical scenarios where its robustness is applicable, assessing DSMOOTH's performance in those contexts.
>
> A: Thank you for pointing this out. We are happy to improve our submission, by adding a clear discussion on this point. Our rationale for using the Mahalanobis distance was aligned with related work,, e.g., [1]. Standard p-norm certificates represent a worst-case scenario., since they constraint the certificate to the p-closest adversary. However, the decision boundaries of general classifiers may be complex and nonlinear, and standard $l_p$ norms may be uninformative in terms of the shape of decision boundaries. On the other hand, the Mahalanobis distance is defined relative to the distribution of the adversarial perturbations, using the covariance matrix to adjust for the spread and correlations of the perturbations. With this submission, our goal was to show that the proposed distance allows us to derive certificates that are more informative w.r.t. the decision boundary. We believe that practical scenarios where this framework is applicable are camera-based smart vision systems, where physical adversarial attacks can successfully mislead perception models. We understand, however, that our work at this stage is insufficient to support these claims.
>
> [1] ANCER: Anisotropic Certification via Sample-wise Volume Maximization. PMLR 2022

---

### Official Review · Reviewer_fV7Q · 2024-11-01

**Soundness:** 2
**Presentation:** 2
**Contribution:** 3
**Rating:** 3
**Confidence:** 5

**Summary:**

This paper proposes *Dynamic Smoothing* (DSMOOTH), an extension of the randomized smoothing framework aimed at enhancing robustness against complex adversarial attacks. Traditional randomized smoothing methods rely on isotropic Gaussian noise, which limits their effectiveness against multi-norm and structured adversarial threats. Authors overcome these limitations by employing a broader range of noise distributions and using Mahalanobis distance to define probabilistic robustness guarantees, making it more adaptable to localized and non-uniform attacks. The authors validate DSMOOTH’s effectiveness through experiments on CIFAR-10 and IMAGENET, where it demonstrates significantly improved certified accuracy against state-of-the-art baselines in multi-attack scenarios.

**Strengths:**

The core strength of the paper is its strong theoretical foundation, presenting a novel idea of using Mahalanobis distance to extend the randomized smoothing framework, enabling it to handle complex adversarial attacks. It demonstrates originality by expanding randomized smoothing to incorporate a range of noise distributions beyond isotropic Gaussian noise, thereby providing robustness against multi-norm, multi-type adversarial threats, which traditional smoothing methods struggle to defend against. This is a valuable and important direction to explore in the context of randomized smoothing. The experimental results also support the theoretical claims, further adding to the strength of the paper.

**Weaknesses:**

While the paper presents promising theoretical advancements, several issues weaken its overall contribution and lead me to lean towards rejection. Although the experimental results support the theoretical claims, the setup itself lacks rigor. For instance, the evaluation is conducted on only 500 images from the CIFAR-10 test set rather than the full test set, potentially skewing the robustness results and limiting the generalizability of the findings. This choice is not well-justified, especially given the scale of CIFAR-10 and the availability of complete test sets. Additionally, the authors choose not to compare against baselines designed for $l_{p}$-norm defenses with $p > 2$, but the reasons for this omission are vague. This exclusion undermines the evaluation since these defenses are common benchmarks in adversarial robustness research. Further, while the paper aims to address robustness in a multi-attack setting, it only tests a single combination of attacks (Square Attack + FGSM). The limitation to just one multi-attack scenario significantly weakens the experimental results, as the approach’s effectiveness under diverse, real-world multi-attack combinations remains unverified. Expanding the experimental setup to include multiple combinations and comprehensive baseline comparisons would strengthen the paper’s contributions.

One more thing I would like to point out is that, the authors discuss isotropic gaussian distribution (this has been referred to as isotopic consistently in the paper which is wrong terminology), something similar has been discussed in few other works in the context of randomized smoothing [1,2], similaritities and/or dissimilarities with those methods is not discussed in the related works section.

References:
[1] Hanbin Hong and Yuan Hong. Certified adversarial robustness via anisotropic randomized smoothing. arXiv preprint arXiv:2207.05327, 2022.
[2] Francisco Eiras, Motasem Alfarra, M Pawan Kumar, Philip HS Torr, Puneet K Dokania, Bernard Ghanem, and Adel Bibi. Ancer: Anisotropic certification via sample-wise volume maximization. arXiv preprint arXiv:2107.04570, 2021.

**Questions:**

1. Could the authors provide the evaluation results of their method on the entire CIFAR-10 test set? Using the full test set is standard in the field and would provide stronger evidence for the method's generalizability.


2. The authors exclude baselines based on $l_{p}$-norm defenses with $p > 2$. Could the authors provide a more detailed justification for this choice? Although this is briefly mentioned in Section 5.1 under **Baselines**, it is not very clear to the reader why this choice was made. Specifically, could the authors expand on the line: *“We do not consider randomized smoothing techniques with certification guarantees in terms of $l_{p}$ norms with $p > 2$, since impossibility results are known for increasing $p$.”*?


3. Under the multi-attack setting, only one combination of attacks (Square Attack + FGSM) was evaluated. Testing multiple attack combinations could better demonstrate DSMOOTH's robustness. Are there plans to expand the experimental setup to include additional attack combinations?


4. Other works have discussed anisotropic approaches in the context of randomized smoothing, such as Hong and Hong (2022) and Eiras et al. (2021). Could the authors elaborate on the similarities or differences between their approach and these methods, particularly in the related works section?

---

> ### Author Response · Authors · 2024-11-22
>
> Q: Could the authors provide the evaluation results of their method on the entire CIFAR-10 test set? Using the full test set is standard in the field and would provide stronger evidence for the method's generalizability.
>
> A: Our experiments are indeed carried out on the entire CIFAR-10 dataset, as in previous related work. Lines 398-400 are incorrect, and they will be removed from the final version of this submission.
>
> Q: The authors exclude baselines based on $p$-norm defenses with $p>2$. Could the authors provide a more detailed justification for this choice? Although this is briefly mentioned in Section 5.1 under Baselines, it is not very clear to the reader why this choice was made. Specifically, could the authors expand on the line: “We do not consider randomized smoothing techniques with certification guarantees in terms of $l_p$ norms with $p>2$, since impossibility results are known for increasing $p$”?
>
> A: This statement refers to Corollary 7.4 by [3], showing that, for any smoothing scheme satisfying Def. 71., the largest possible certified radius decreases w.r.t. $p$. We understand that this argument is vague and that additional discussion is needed.
>
> Q: Under the multi-attack setting, only one combination of attacks (Square Attack + FGSM) was evaluated. Testing multiple attack combinations could better demonstrate DSMOOTH's robustness. Are there plans to expand the experimental setup to include additional attack combinations?
>
> A: Thank you for raising this point. We are happy to incorporate more attacks (or attack combinations) in a future re-submission.
>
> Q: Other works have discussed anisotropic approaches in the context of randomized smoothing, such as Hong and Hong (2022) and Eiras et al. (2021). Could the authors elaborate on the similarities or differences between their approach and these methods, particularly in the related works section?
>
> A: Hong and Hong (2022): The certification guarantees in this work (Hong and Hong (2022), Thm. 4.1) provide guarantees for the case where the smoothing distribution is of the form N(0, Sigma), where Sigma is a diagonal matrix. These guarantees are different from our work, which derives probabilistic guarantees for any positive-definite matrix Sigma. Furthermore, Hong and Hong (2022) differs from our work in the way that the matrix Sigma is chosen. Eiras et al. (2021) differs from our work in the way that the matrix Sigma is obtained. In Eiras et al. (2021), the matrix Sigma is computed by solving an optimization problem as in Eq. (1).
>
> [1] Hanbin Hong and Yuan Hong. Certified adversarial robustness via anisotropic randomized smoothing. arXiv preprint arXiv:2207.05327, 2022.
> [2] Francisco Eiras, Motasem Alfarra, M Pawan Kumar, Philip HS Torr, Puneet K Dokania, Bernard Ghanem, and Adel Bibi. Ancer: Anisotropic certification via sample-wise volume maximization. arXiv preprint arXiv:2107.04570, 2021.
> [3] Greg Yang, et. al. Randomized Smoothing of All Shapes and Sizes, ICML 2020

---

### Note · Authors · 2024-11-22

**Comment:**

We would like to thank the reviewers for taking the time to provide insightful feedback. We understand that the current submission has various shortcomings, including lack of novelty, as well as the other issues pointed out by the reviewers. For this reason, we have decided to withdraw this submission.

**Withdrawal Confirmation:**

I have read and agree with the venue's withdrawal policy on behalf of myself and my co-authors.